# Small Nash Equilibrium Certificates in Very Large Games

**Brian Hu Zhang**
Computer Science Department
Carnegie Mellon University
bhzhang@cs.cmu.edu

**Tuomas Sandholm**
Computer Science Department, CMU
Strategic Machine, Inc.
Strategy Robot, Inc.
Optimized Markets, Inc.
sandholm@cs.cmu.edu

## Abstract

In many game settings, the game is not explicitly given but is only accessible by playing it. While there have been impressive demonstrations in such settings, prior techniques have not offered safety guarantees, that is, guarantees on the game-theoretic exploitability of the computed strategies. In this paper we introduce an approach that shows that it is possible to provide exploitability guarantees in such settings without ever exploring the entire game. We introduce a notion of a *certificate* of an extensive-form approximate Nash equilibrium. For verifying a certificate, we give an algorithm that runs in time linear in the size of the certificate rather than the size of the whole game. In zero-sum games, we further show that an optimal certificate—given the exploration so far—can be computed with any standard game-solving algorithm (e.g., using a linear program or counterfactual regret minimization). However, unlike in the cases of normal form or perfect information, we show that certain families of extensive-form games do not have small approximate certificates, even after making extremely nice assumptions on the structure of the game. Despite this difficulty, we find experimentally that very small certificates, even exact ones, often exist in large and even in infinite games. Overall, our approach enables one to try one's favorite exploration strategies while offering exploitability guarantees, thereby decoupling the exploration strategy from the equilibrium-finding process.

## 1 Introduction

Recent years have witnessed AI breakthroughs in games such as poker [5, 28, 10, 12] where the rules are given. In many important applications—such as many war games and finance simulations—the rules are only given via *black-box* access, that is, via playing the game [36, 25], and one can try to construct good strategies by self play. In such settings, deep reinforcement learning techniques are typically used today [16, 33, 25, 34, 35, 2]. However, such methods lack the guarantee of low (or zero) *exploitability* that game-theoretic solving techniques offer.

Prior to our paper, to compute exploitability of a strategy, one needed to compute the other player's best response to it, which relies on the game being known. Sampling approaches to equilibrium finding have been suggested, but their regret guarantees are vacuous unless the algorithms touch at least as many information sets as there are in the game [24, 34, 38]. A recent PAC-learning algorithm has logarithmic sample complexity for *pure* maxmin strategies in *normal-form* games; it extends to some infinite games, but not effectively to mixed strategies in extensive-form games [27].

*Game abstraction* is commonly used to reduce the size of a game tree prior to solving [3, 14, 8, 13]. Practical abstraction techniques were fundamental to achieving superhuman performance in no-limit

Texas hold'em poker in the *Libratus* [10] and *Pluribus* [12] agents. However, these techniques do not have exploitability guarantees. There has been recent work on abstraction algorithms with exploitability guarantees for specific settings [31, 1] and for general extensive-form games (e.g., [21, 22]), but these are not scalable for large games such as no-limit Texas hold'em, and the guarantees depend on the difference between the abstracted game and the real game being known.

We introduce an approach that can provide exploitability guarantees (even zero exploitability) in black-box games without ever exploring the entire game tree. We introduce a notion of certificate that is often much smaller than the full game. We show that a certificate can be verified in time linear in the size of the certificate, without expanding the remainder of the game tree. For zero-sum games, we give an algorithm that computes an optimal certificate given the current set of explored nodes using any zero-sum game solver as a subroutine. Leveraging prior results, we show that perfect-information [19] and normal-form [26] games have short certificates. We prove that extensive-form games do not always have such, but under a certain informational assumption they do. We also show that it is NP-hard to approximate to within a logarithmic factor the smallest certificate of a game, even in the zero-sum setting, and give an exponential lower bound for the time complexity of solving a black-box game as a function of the size of its smallest certificate. Despite these hardness results, we give a game-solving algorithm that expands nodes incrementally until a certificate is found. It often terminates while only exploring a small fraction of the tree, and works even when the game tree is infinite and payoffs may be unbounded. Our experiments show that large and even infinite games can be solved exactly while expanding only a small fraction of the game tree.

## 2 Preliminaries

We study *extensive-form games*, hereafter simply *games*. An extensive-form game consists of the following:

(1) a set of players $\mathcal{P}$, usually identified with positive integers $1, 2, \ldots n$. *Nature*, a.k.a. *chance*, will be referred to as player 0. For a given player $i$, we will often use $-i$ to denote all players except $i$ and nature.

(2) a finite tree $H$ of *histories*, rooted at some *initial state* $\emptyset \in H$. The set of leaves, or *terminal states*, in $H$ will be denoted $Z$. The edges connecting any node $h \in H$ to its children are labeled with *actions*.

(3) a map $P : H \to \mathcal{P} \cup \{0\}$, where $P(h)$ is the player who acts at node $h$ (possibly nature).

(4) for each player $i$, a *utility function* $u_i : Z \to \mathbb{R}$. 5) for each player $i$, a partition of player $i$'s decision points, i.e., $P^{-1}(i)$, into *information sets*. In each information set $I$, every pair of nodes $h, h' \in I$ must have the same set of actions. 6) for each node $h$ at which nature acts, a distribution $\sigma_0(\cdot|h)$ over the actions available to nature at node $h$.

We will use $(G, u)$, or simply $G$ when the utility function is clear, to denote a game. $G$ contains the tree and information set structure, and $u = (u_1, \ldots, u_n)$ is the profile of utility functions. For any history $h \in H$ and any player $i \in \mathcal{P}$, the *sequence* of player $i$ at node $h$ is the sequence of information sets observed and actions taken by player $i$ on the path from the root node to $h$. In this paper, all games are assumed to have perfect recall.

A *behavior strategy* (hereafter simply *strategy*) $\sigma_i$ for player $i$ is, for each information set $I \in J_i$ at which player $i$ acts, a distribution $\sigma_i(\cdot|I)$ over the actions available at that infoset. When an agent reaches information set $I$, it chooses action $a$ with probability $\sigma_i(a|I)$.

A collection $\sigma = (\sigma_1, \ldots, \sigma_n)$ of behavior strategies, one for each player $i \in \mathcal{P}$, is a *strategy profile*. The *reach probability* $\sigma_i(h)$ is the probability that node $h$ will be reached, assuming that player $i$ plays according to strategy $\sigma_i$, and all other players (including nature) always choose actions leading to $h$ when possible. Analogously, we define $\sigma(h) = \prod_{i \in \mathcal{P} \cup \{0\}} \sigma_i(h)$ to be the probability that $h$ is reached under strategy profile $\sigma$. This definition naturally extends to sets of nodes or to sequences by summing the reach probabilities of all relevant nodes. A strategy profile induces a distribution over the terminal nodes of the game. The *value* of a strategy profile $\sigma$ for player $i$ is $u_i(\sigma) := \mathbb{E}_{z \sim \sigma} u_i(z)$.

The *best response value* $u_i^*(\sigma_{-i})$ for player $i$ against an opponent strategy $\sigma_{-i}$ is the largest achievable value; i.e. in a two-player game, $u_i^*(\sigma_{-i}) = \max_{\sigma_i} u_i(\sigma_i, \sigma_{-i})$. A strategy $\sigma_i$ is an $\varepsilon$-*best response* to opponent strategy $\sigma_{-i}$ if $u_i(\sigma_i, \sigma_{-i}) \geq u_i^*(\sigma_{-i}) - \varepsilon$.

A strategy profile $\sigma$ is an $\varepsilon$-*Nash equilibrium* (NE) if all players are playing $\varepsilon$-best responses. *Best responses* and *Nash equilibria* are respectively 0-best responses and 0-Nash equilibria.

## 3 $\varepsilon$-Nash certificates via pseudogames

We are interested in finding small *certificates* of exact and approximate Nash equilibria. We introduce a construct that we call a *pseudogame*, which can be used to build small certificates of equilibria.

**Definition 3.1.** A *pseudogame* $\tilde{G} = (\tilde{G}, \alpha, \beta)$ is a game in which some terminal nodes do not have specified utility but rather have only lower and upper bounds on utilities. Formally, for each player $i$, instead of the standard utility function $u_i : Z \to \mathbb{R}$, there are lower and upper bound functions $\alpha_i : Z \to \mathbb{R}$ and $\beta_i : Z \to \mathbb{R}$ indicating lower and upper bounds respectively on the utility of a node. We demand $\alpha_i(z) \le \beta_i(z)$ for every $i$ and $z$. We call a node *pseudoterminal* if $\alpha_i(z) < \beta_i(z)$ for some $i$, and use *terminal node* to refer to any leaf in a pseudogame.

**Definition 3.2.** An $\varepsilon$-Nash equilibrium of a pseudogame $(\tilde{G}, \alpha, \beta)$ is a strategy profile $\sigma$ for which, for every player $i$, we have $\beta_i^*(\sigma_{-i}) - \alpha_i(\sigma) \le \varepsilon$.

**Definition 3.3.** A pseudogame $(\tilde{G}, \alpha, \beta)$ is a *trunk* of a game $(G, u)$ if:

(1) $\tilde{G}$ can be created by collapsing some internal nodes of $G$ into terminal nodes (and removing them from information sets they are contained in), and

(2) if $h$ is a pseudoterminal node of $\tilde{G}$, and $z$ is a terminal node of $G$ that is a descendant of $h$, then $\alpha_i(h) \le u_i(z) \le \beta_i(h)$ for every $i$. That is, the bounds $\alpha$ and $\beta$ are correct.

It is possible for information sets of a game $G$ to be partially or totally removed in a trunk game.

**Definition 3.4.** An $\varepsilon$-*certificate* for a game $G$ is a pair $(\tilde{G}, \sigma)$, where $\tilde{G}$ is a trunk of $G$ and $\sigma$ is an $\varepsilon$-Nash equilibrium of $\tilde{G}$.

Importantly, the definition of a certificate is independent of the original game $G$; that is, given $(\tilde{G}, \sigma^*)$, $\varepsilon$ can be computed without knowing the remainder of the game tree of $G$: by computing the best response for each player in their optimistic game, it can be done in time linear in the size of $\tilde{G}$.

The proposition below shows that our definition of certificate is reasonable. Proofs are in the appendix.

**Proposition 3.5.** *Let $(\tilde{G}, \sigma)$ be an $\varepsilon$-certificate for game $G$. Then any strategy profile in $G$ created by playing according to $\sigma$ in any information set appearing in $\tilde{G}$ and arbitrarily at information sets not appearing in $\tilde{G}$ is an $\varepsilon$-NE in $G$.*

## 4 Do small certificates exist?

In this section, we study when games have small $\varepsilon$-certificates. Our general goal will be to find certificates of size $O(N^c \operatorname{poly}(1/\varepsilon))$ for some universal constant $c < 1$, where $N$ is the number of nodes. If a game has a small certificate, there is hope of finding such a certificate quickly, and thus being able to find and verify an (approximate or exact) Nash equilibrium while exploring only a small part of the game. We start by giving a connection between sparse equilibria and small certificates, which we will use later in this section.

**Proposition 4.1** (Sparse equilibria imply small certificates)**.** *Let $\sigma$ be an $\varepsilon$-NE of a game $G$, and let $\tilde{G}$ be the smallest trunk of game $G$ containing every node $h$ for which $\sigma_{-i}(h) > 0$ for any player $i$. Then $(\tilde{G}, \sigma)$ is an $\varepsilon$-certificate of $G$.*

### 4.1 Perfect-information zero-sum games have small certificates, via alpha-beta search

In two-player perfect-information zero-sum games, under certain assumptions, small certificates exist. Specifically, assume that

(1) there is no randomness (no nature nodes),

(2) all nodes have uniform branching factor $b = O(1)$,

(3) moves alternate; i.e., a player-1 decision node is always followed by a player-2 decision node, and

(4) the tree has uniform depth $d$.

In this case, the game has $N = b^d$ terminal nodes. Alpha-beta search with an optimal heuristic will search only $O(b^{d/2}) = O(\sqrt{N})$ tree nodes before arriving at a provably optimal strategy [19]. Thus, the portion of the game tree consisting of nodes touched by alpha-beta search contains $O(\sqrt{N})$ nodes, and constitutes a 0-certificate.

## 4.2 Normal-form games have small certificates, via sparse equilibria

A *normal-form game* is a game in which each player has only a single information set. A two-player normal-form game with $a_1$ player-1 moves and $a_2$ player-2 moves (hence $N = a_1 a_2$ terminal nodes) can thus be expressed as a pair of *utility matrices* $A, B \in \mathbb{R}^{a_1 \times a_2}$. In two-player normal-form games, for every $\varepsilon$, there is an $\varepsilon$-NE in which each player $i$ randomizes over $O(\log(a_{-i})/\varepsilon^2)$ pure strategies [26]. Let $\sigma^*$ be such an $\varepsilon$-Nash equilibrium, and let $S_i \subseteq [a_i]$ be the support of $\sigma_i$.

Consider the following extensive-form pseudogame: First, P1 chooses her strategy $s_1 \in [a_1]$. Then, P2 decides whether or not she should play a node from $S_2$. If P2 decides not to play from $S_2$, and P1 has not played an action in $S_1$, the pseudogame terminates immediately in a pseudoterminal node with trivial payoff bounds, i.e., $(-\infty, \infty)$. Otherwise, P2 chooses some strategy $s_2 \in S_2$ to play, and the proper payoffs are given out. This pseudogame has $O(a_1|S_2| + a_2|S_1|)$ terminal nodes, and by Proposition 4.1, the profile $\sigma^*$ is an $\varepsilon$-NE in it. Thus, when $a_1 = \Theta(a_2)$, an $a_1 \times a_2$ normal-form game has an $\varepsilon$-certificate of size $O(\sqrt{N} \log(N)/\varepsilon^2)$.

Unlike in the case of perfect-information zero-sum games, normal-form games in general do not have small *exact* certificates: an exact certificate must necessarily include all strategies played in some equilibrium, and there are normal-form games for which the only equilibria are fully mixed.

## 4.3 Extensive-form games with low information have small certificates

This can be generalized to extensive-form games where players do not learn too much information.

**Theorem 4.2.** *Let $G$ be a two-player game with $N$ nodes and bounded payoffs, and let $D$ be the maximum number of terminal sequences in the support of any pure strategy for either player. Then $G$ has an $\varepsilon$-Nash equilibrium in which both players mix among $O((D^2/2\varepsilon^2) \log N)$ pure strategies.*

Intuitively, $D$ is a measure of how much information the players have in the game. A player who learns no information whatsoever throughout the game will have $D = 1$, so this proposition matches the sparseness result [26] in the normal-form case. On the other hand, a player with perfect information may have $D = \Omega(\sqrt{N})$ or even larger, in which case this proposition is vacuous.

Under the assumptions of Section 4.1 except perfect information, any given pure strategy is supported on $O(\sqrt{N})$ nodes. Thus, by Proposition 4.1, we have the following result which implies the existence of small certificates when $D = O(N^c)$ for $c < 1/4$:

**Corollary 4.3.** *Under the assumptions of Theorem 4.2 and Section 4.1 except perfect information, $G$ has an $\varepsilon$-certificate of size $O(\sqrt{N}(D^2/\varepsilon^2) \log N)$.*

As in the case of normal-form games, in general, exact certificates may need to include the whole game tree. However, in some cases, we can do better. For example, games with a natural *public game tree*[1] [18] often have sparse equilibrium strategies [32] and thus small certificates by Proposition 4.1. We will also show later with empirical experiments that many practical games have small exact certificates.

## 4.4 Small certificates do not always exist in extensive-form games

In light of the above results, one might hope that there are sparse approximate equilibria in extensive-form games, which would allow small certificates in such games:

**Question 4.4** (Existence of small $\varepsilon$-certificates). *Let $G$ be a two-player zero-sum game with $N$ nodes. Suppose that $G$ satisfies the assumptions in Section 4.1. Let $\varepsilon > 0$. Is there always an $\varepsilon$-certificate with $O(N^c \operatorname{poly}(1/\varepsilon))$ tree nodes, for some universal constant $c < 1$?*

It would be nice if this had a positive answer, since that would interpolate between the cases of normal form and perfect information, which, as discussed above, both have $\tilde{O}(\sqrt{N}/\varepsilon^2)$-sized certificates. We show that, unfortunately, the answer is negative. As a counterexample, consider playing $T$ rounds of matching pennies. After each round, P2 learns what P1 played, but P1 does not learn what P2 played. Each round is worth $1/T$ points, so the maximum score is 1. The game tree has uniform depth $2T$ and uniform branching factor 2, for a total of $N := 2^{2T}$ terminal nodes.

**Theorem 4.5.** *Any $\varepsilon$-certificate of this game must have at least $\Omega(N^{1-O(\varepsilon)})$ nodes.*

It does not help to add the assumption that the game is win-loss: any zero-sum game can be made win-loss by adding normal-form gadget games to the terminal nodes which force the players to mix.

# 5 Black-box setting

For the remainder of this paper, we will assume that we are not given access to the full game tree. Instead, we are only given black-box access to the game, in the form of a function that, given a node $h$ (in the form of a history of actions), gives us:

(1) upper and lower bounds on the value of any terminal descendant of $h$,

(2) if $h$ is nonterminal, the player to act at that node, and a list of legal actions; and

(3) if the player to act at $h$ is nature, a single sampled action from nature's action distribution.

The game may possibly be very large, or even infinite, but we will assume that every node has some terminal descendant (so that (1) is well-defined), and that the game has a finite $0$-certificate. The bounds given by (1) may be infinite, either because the oracle does not give optimal bounds, or because the game is infinite and the payoffs along a branch may be unbounded.

The first challenge is approximating the true nature distributions via samples. We thus give a result regarding the sample complexity of doing this for a given pseudogame with bounded payoffs[2].

**Theorem 5.1** (Sample complexity of approximating a game). *Let $G$ be a game with $N$ nodes and bounded payoffs, and suppose that the true nature distributions are unknown but have been approximated by sampling at every nature node. Let $\hat{\sigma}_0$ be the approximated nature strategy resulting from this sampling. Fix a player $i$. Let $\hat{u}_i(\sigma)$ denote the expected utility of player $i$ when the players play strategy $\sigma$ and nature plays $\hat{\sigma}_0$. Let $D$ be the maximum support size over terminal nodes of any pure strategy profile in the perfect-information refinement of $G$. Suppose that, for every nature node $h$ is sampled at least $\hat{\sigma}_0(h)(D^2/2\varepsilon^2)\log(2N/\delta)$ times. Then, with probability $1 - \delta$, for any strategy profile $\sigma$, we have $|u_i(\sigma) - \hat{u}_i(\sigma)| \leq \varepsilon$.*

Here, $D$ is some measure of how much randomness there is in $G$. For example, if $G$ has no nature nodes, $D = 1$. If $G$ has no player nodes, $D = N$.

**Corollary 5.2.** *Let $\tilde{G}$ be a pseudogame, and consider approximating nature's strategy in $\tilde{G}$ to precision $\varepsilon$ as per Theorem 5.1. Let $\sigma$ be an $\varepsilon'$-equilibrium of the approximated version of $\tilde{G}$. Then $\sigma$ is also an $(\varepsilon' + 2\varepsilon)$-equilibrium of $\tilde{G}$ with probability at least $1 - 2\delta|\mathcal{P}|$.*

In the above results, the (pseudo)game and sample size at each nature node $h$ are both held fixed; the probability is only over the random samples themselves. Thus, if running an algorithm that incrementally expands nodes in a pseudogame, the samples should in principle be re-drawn every time $\tilde{G}$ changes. The factor of $2|\mathcal{P}|$ is not bothersome since $|\mathcal{P}| \leq N$ surely, so this incurs at most a constant factor in the sample complexity. Importantly, the sample complexity depends only on the size and structure of the pseudogame $\tilde{G}$, not on whatever full game $G$ that $\tilde{G}$ may be a trunk of.

In the rest of the paper, both for simplicity and to allow discussion of the case of unbounded payoffs, we will not deal with sampling. Instead, we will assume that the exact nature action distribution is given by the black-box oracle when a nature node is reached.

# 6 The zero-sum case

Our results so far have been valid for $n$-player general-sum games unless otherwise stated. In this section we focus on two-player zero-sum games, where one can hope[3] to perhaps efficiently *find* small certificates. A two-player game is *zero-sum* if $u_1 = -u_2$. In this case, we refer to a single utility function $u$; it is understood that player 2's utility function is $-u$. In zero-sum games, all Nash equilibria have the same expected value; this is called the *value of the game*, and we denote it by $u^*$. The *exploitability* of an opponent strategy $\sigma_{-i}$ for player $i$ is then $|u^*(\sigma_{-i}) - u^*|$.

## 6.1 Certificates in zero-sum games

In the zero-sum case, we use a slightly different notion of $\varepsilon$-equilibrium of a pseudogame, which will make the subsequent results more precise.

**Definition 6.1.** A two-player pseudogame $(\tilde{G}, \alpha, \beta)$ is zero-sum if $\alpha_2 = -\beta_1$ and $\beta_2 = -\alpha_1$.

As alluded to above, in this situation, we will drop the subscripts, and write $\alpha$ and $\beta$ to mean $\alpha_1$ and $\beta_1$. In particular, $(\tilde{G}, \alpha)$ and $(\tilde{G}, \beta)$ are zero-sum games.

**Definition 6.2.** An $\varepsilon$-Nash equilibrium of a two-player zero-sum pseudogame $(\tilde{G}, \alpha, \beta)$ is a strategy profile $(x^*, y^*)$ for which $\beta^*(y^*) - \alpha^*(x^*) \leq \varepsilon$.

In this sense, $\varepsilon$ is the sum of the exploitabilities of both players' strategies. These are related to Definition 3.2 as follows:

**Proposition 6.3.** *Any $\varepsilon$-NE in the sense of Definition 6.2 is an $\varepsilon$-NE in the sense of Definition 3.2.*

**Proposition 6.4.** *Any $\varepsilon$-NE in the sense of Definition 3.2 is a $2\varepsilon$-NE in the sense of Definition 6.2.*

Let $(\tilde{G}, \alpha, \beta)$ be a pseudogame. Let $(x_*, y_*)$ be a Nash equilibrium of the game $(\tilde{G}, \alpha)$, and $(x^*, y^*)$ be a Nash equilibrium of $(\tilde{G}, \beta)$. We will call the pair of strategies $(x_*, y^*)$ a *pessimistic equilibrium* of $(\tilde{G}, \alpha, \beta)$ since both players are playing as if their utilities are as bad as possible. Similarly, we will call $(x^*, y_*)$ an *optimistic profile*[4].

By definition, the pessimistic equilibrium is an $\varepsilon$-NE of $(\tilde{G}, \alpha, \beta)$, where $\varepsilon = \beta^* - \alpha^*$. This gives us an algorithm for finding the best certificate from a given trunk, that runs in time polynomial in the size of the trunk: to get a strategy for P1 (the maximizer player), solve the game $(\tilde{G}, \alpha)$, and to get a strategy for P2, solve $(\tilde{G}, \beta)$. Since the zero-sum game solver is used strictly as a subroutine, any solver of choice may be used: for example, a linear program (LP) solver with the sequence-form LP [20, 37], modern variants of CFR [11, 9, 6, 7], or first-order methods [17, 23]. If the solver only finds an $\varepsilon'$-equilibrium of the game it is solving, the result is a certificate for $(\varepsilon + 2\varepsilon')$-equilibrium.

## 6.2 Lower bounds

Since solving zero-sum games can be done efficiently, there is some hope that small certificates can also be found efficiently. Another goal may be to find a certificate efficiently, say, in time polynomial in the size of the smallest certificate of a given game. Unfortunately, these are both impossible:

**Theorem 6.5** (Hardness of approximating the smallest certificate). *Assuming $P \neq NP$, there is no $\mathrm{poly}(N, 1/\varepsilon)$-time algorithm that, given the game tree of a zero-sum game with $N$ nodes, outputs the smallest $\varepsilon$-certificate of the game to better than a $\Theta(\log N)$ factor of approximation.*

**Theorem 6.6.** *There is no algorithm for zero-sum game solving in the black-box setting, even assuming bounded branching factor, with runtime subexponential in the size of the smallest certificate.*

These hardness results have slightly different flavors and consequences. The hardness in Theorem 6.5 comes from the imperfect information: in the perfect-information setting, the task can be done with a variant of alpha-beta search in linear time. Further, in practice, we usually do not care about finding

the *smallest* certificate, as long as we can efficiently find one of reasonable size. The hardness in Theorem 6.6 is more fundamental: it comes from the fact that we cannot assume access to any reasonable heuristic of where to explore; thus, we may explore the optimal path of play last in the worst case, resulting in a large certificate.

## 6.3 An algorithm for solving black-box games

Despite the difficulties presented by Theorems 6.5 and 6.6, we present an algorithm for finding a certificate in a zero-sum game in the black-box setting, with nontrivial provable guarantees. For now, we will assume that the game $\tilde{G}$ has bounded payoffs; later we will relax this assumption.

---

**Algorithm 6.7** Finding a certificate in a two-player zero-sum game

1: start with a pseudogame $(\tilde{G}, \alpha, \beta)$ that has only a root node.
2: **loop**
3:     solve $(\tilde{G}, \alpha)$ and $(\tilde{G}, \beta)$ with an LP solver to obtain equilibria $(x_*, y_*)$ and $(x^*, y^*)$.
4:     expand all pseudoterminal nodes of $\tilde{G}$ that appear in the support of $(x^*, y_*)$.
5:         (if there are none, stop and output $\tilde{G}$ and the pessimistic equilibrium $(x_*, y^*)$.)

---

We use LP for the game solves in Line 3, for three reasons. First, LP[5] results in an exact solution (at least up to numerical tolerances), which is desirable because the support of the solution is relevant to Line 4; iterative solvers such as CFR typically return fully mixed solutions. Second, only a small number of changes are made to the LP with each node expanded, so LP algorithms that can be warm started, such as primal or dual simplex, can be efficient in practice. Third, it will allow us to adapt this algorithm to the case of unbounded payoffs, which we will see later; again, CFR cannot do that.

From the discussion in Section 6.1, we know that this algorithm will always output an 0-certificate. If we want an $\varepsilon$-certificate for $\varepsilon > 0$, we can also simply terminate the algorithm when $\beta^* - \alpha^* < \varepsilon$. We now prove an important fact about Algorithm 6.7.

**Theorem 6.8.** *A pseudogame has a* 0-*Nash equilibrium if and only if it has an optimistic profile with no pseudoterminal node in its support.*

The "only if" direction guarantees that Line 4 does not terminate the algorithm unless a 0-certificate has been found. The "if" direction guarantees a weak form of "this algorithm will not waste work": modulo the uniqueness of the optimistic profile[6], the algorithm stops exactly when it has found a 0-certificate. This is not trivial: other protocols such as "expand all pseudoterminal nodes appearing in the support of at least one player in the pessimistic equilibrium" fail to satisfy the "if" direction.

The algorithm has no runtime bound as a function of the size of the smallest certificate of $G$, even assuming bounded branching factor: indeed, if $G$ is infinite, it is even possible for the algorithm to run indefinitely, even when a finite-sized certificate exists. One way to fix this without losing more than a constant factor in efficiency is to, in addition to Line 4, also always expand the shallowest strictly pseudoterminal node of $\tilde{G}$ at each iteration. This way, a certificate with $d$ nodes has depth at most $d$, and thus will be generated after at most after $O(b^d)$ expansions (where $b$ is a bound on the branching factor of the game), matching the lower bound of Theorem 6.6.

## 6.4 Handling unbounded payoffs

In infinite games with unbounded payoffs, it is possible for the games $(\tilde{G}, \alpha)$ and $(\tilde{G}, \beta)$ to have infinite-magnitude utility on some nodes. For example, $(\tilde{G}, \beta)$ may have payoff $+\infty$ on some nodes (but not $-\infty$). We now show how to adapt Algorithm 6.7 for such situations. Assume WLOG that we are solving $(\tilde{G}, \beta)$; i.e. it is possible for payoffs to be $+\infty$ but not $-\infty$ (for $(\tilde{G}, \alpha)$, swap the players). Call a P2-sequence *bad* if its support (over terminal nodes) contains a node of utility $+\infty$. Assume that it is possible for P2 to avoid all bad sequences; otherwise, the game has value $+\infty$. Consider the

Table 1: Experimental results. The *minimal certificate* is a certificate after removing all unnecessary nodes per Proposition 4.1. Percentages are relative to game size. Leduc variants have infinite size; for them, "game size" reported is for the trunk with the number of consecutive raises restricted to 12.

| game | size of game | | size of certificate | | | | size of minimal certificate | | | |
|---|---|---|---|---|---|---|---|---|---|---|
| | nodes | infosets | nodes | | infosets | | nodes | | infosets | |
| search game | 234,705 | 11,890 | 13,682 | 5.8% | 532 | 4.5% | 5,526 | 2.4% | 379 | 3.2% |
| 4-rank PI Goofspiel | 2,229 | 1,653 | 275 | 12.3% | 110 | 6.7% | 141 | 6.3% | 54 | 3.3% |
| 5-rank PI Goofspiel | 55,731 | 41,331 | 2,593 | 4.7% | 957 | 2.3% | 763 | 1.4% | 288 | 0.7% |
| 6-rank PI Goofspiel | 2,006,323 | 1,487,923 | 21,948 | 1.1% | 7,584 | 0.5% | 4,438 | 0.2% | 1,677 | 0.1% |
| 4-rank Goofspiel | 2,229 | 738 | 614 | 27.5% | 117 | 15.9% | 294 | 13.2% | 58 | 7.9% |
| 5-rank Goofspiel | 55,731 | 9,948 | 11,415 | 20.5% | 2,160 | 21.7% | 8,518 | 15.3% | 1,792 | 18.0% |
| 6-rank Goofspiel | 2,006,323 | 166,002 | 266,756 | 13.3% | 15,776 | 9.5% | 171,343 | 8.5% | 12,135 | 7.3% |
| 3-rank random Goofspiel | 1,066 | 426 | 309 | 29.0% | 92 | 21.6% | 214 | 20.1% | 65 | 15.3% |
| 4-rank random Goofspiel | 68,245 | 17,432 | 16,416 | 24.1% | 3,270 | 18.8% | 11,992 | 17.6% | 2,335 | 13.4% |
| 5-rank random Goofspiel | 8,530,656 | 1,175,330 | 1,854,858 | 21.7% | 241,985 | 20.6% | 1,388,172 | 16.3% | 185,946 | 15.8% |
| 5-rank limit Leduc | *197,736* | *13,920* | 26,306 | *13.3%* | 2,406 | *17.3%* | 12,923 | *6.5%* | 1,242 | *8.9%* |
| 9-rank limit Leduc | *1,181,512* | *44,928* | 137,662 | *11.7%* | 6,811 | *15.2%* | 51,533 | *4.4%* | 2,891 | *6.4%* |
| 13-rank limit Leduc | *3,578,472* | *93,600* | 337,312 | *9.4%* | 12,171 | *13.0%* | 105,769 | *3.0%* | 4,449 | *4.8%* |

sequence-form bilinear saddle-point problem [20] for $(\tilde{G}, \beta)$ (left) and its equivalent LP (right):

$$\max_{x \geq 0} \min_{y \geq 0} x^T A y \ \text{ s.t. } \ Bx = b, Cy = c, x, y \geq 0 \qquad \max_{x \geq 0, z} c^T z \ \text{ s.t. } \ Bx = b, C^T z \leq A^T x.$$

Here $A$ is the payoff matrix, which may contain infinite entries. Then, the main idea is to remove any constraint corresponding to bad P2-sequences, and solve the resulting LP (which now by construction contains no infinite entries and is thus well formed), for a Nash equilibrium solution $x$. The problem is that $x$ may not be a true Nash equilibrium of $(\tilde{G}, \beta)$, since it is possible for P1 to end up avoiding nodes of utility $+\infty$, which could allow P2 to best respond by actually playing toward a bad sequence.

Let $V^*(s)$ denote the value that P2 receives by playing a best response to $x$ starting at a P2 infoset or sequence $s$. Let $V(s)$ denote the same, except while forcing P2 to avoid bad sequences. Obviously, $V^* \leq V$. Consider the following recursive algorithm, which we run on every P2-root infoset $I$:

---
**Algorithm 6.9** CORRECT($I$): Correcting a strategy in the case of infinite reward
---
1: **for** each action $a$ available to P2 **do**
2:    **if** $V^*(Ia) < V(I)$ **then**
3:       **for** every P1-sequence $i$ such that $A_{i,Ia} = +\infty$ **do** $x_i \leftarrow x_i + $ infinitesimal[7]
4:       **for** every P2-infoset $I'$ whose parent sequence is $Ia$ **do** CORRECT($I'$)
---

Call a pair of strategies a *corrected optimistic profile* if it is the result of applying this procedure to both parts of an optimistic profile. We can now make the following strengthening of Theorem 6.8:

**Theorem 6.10.** *A pseudogame with possibly unbounded payoffs has a* 0-*Nash equilibrium if and only if it has a* corrected *optimistic profile with no pseudoterminal node in its support.*

Thus, to run Algorithm 6.7 in games with unbounded payoffs, it suffices to apply the correction algorithm to the optimistic profile found in Line 3 before expanding nodes.

# 7 Experiments

We conducted experiments using the algorithm in Section 6 on the following common zero-sum benchmark games.

    (1) A zero-sum variant of the **search game** [4].

(2) $k$-**rank Goofspiel**. It is played as follows. At time $t$ (for $t = 1, \ldots, k$), players place bids for a prize of value $t$. The possible bids are the integers $1, \ldots, k$, and each player must bid each integer exactly once. The player with the higher bid wins the prize; if the bids are equal, the prize is split equally. The winner of each round is made public after each round, but the bids are not. The goal of each player is to maximize the sum of the values of her prizes won. In the *perfect-information* (PI) variant, P2 knows P1's bid while bidding, and bids are made public after each round. This creates a perfect-information game in which P2 has a large advantage, and in which we expect a certificate of size $O(\sqrt{N})$. In the *random* variant, the order of the prizes is randomized.

(3) $k$-**rank limit Leduc poker**. It is a small variant of limit poker, played with one hole card and one community card, and a deck with $k$ ranks. The players are only allowed to raise by a fixed amount, but can do so an unlimited number of times. Thus, the possible payoffs in the game, and the length of the game, are both unbounded.

We computed $0$-certificates in all cases. For the LP solver, we used Gurobi v9.0.0 [15]. Results of experiments can be found in Table 1. In many games, we found $0$-certificates of size substantially smaller than the number of nodes in the game, and the certificate size as a fraction of the game size decreases as the game grows.

The results in Goofspiel align with the theoretical predictions: perfect-information games have very small certificates (basically $\sqrt{N}$ nodes). In light of Proposition 4.1, it also makes sense that certificates are smaller (relative to the size of the game) when there is no randomness: randomness simply increases the number of nodes in the game tree represented by any given pure strategy, so an equilibrium with the same sparsity for the players now leads to a larger certificate.

In Leduc poker, no node involving more than 12 consecutive raises was ever expanded in any size of game while searching for a certificate. This suggests that it is never optimal for either player to play past this point, despite the fact that continuing to raise could in principle lead to an unbounded payoff. This phenomenon allows our algorithm to find a finite-sized $0$-certificate, thus completely solving the game in a reasonably efficient manner, even though it has infinite size.

## 8   Conclusions and future research

We presented a notion of certificate for general extensive-form games that allows verification of exact and approximate Nash equilibria without expanding the whole game tree. We showed that small equilibria exist in some restricted classes of extensive-form game, but not all. We presented algorithms for both verifying a certificate and computing the optimal certificate given the currently-explored trunk of a game. Our experiments showed that many large or even infinite games have small certificates, allowing us to find equilibria while exploring a vanishingly small portion of the game.

This paper opens many directions for future research:

(1) Develop further the ideas of Section 5 for the case of unknown nature distributions. For example, what is the best way to balance sampling, game tree exploration, and equilibrium finding?

(2) Seek algorithms for finding certificates that give stronger guarantees of optimality than Theorem 6.10, especially in the case of infinite games with unbounded utilities.

(3) Seek algorithms with stronger guarantees than that implied by Proposition 4.1 for verifying the Nash gap of a given strategy profile; for example, is it possible to easily construct the smallest trunk for which a given $\sigma$ is an $\varepsilon$-equilibrium?

## Broader Impacts

The techniques have broad applicability. Furthermore, the paper opens up additional important research directions.

Improving the strategic capabilities of people and companies will typically (but not always) improve systemwide good as the players will be able to better reach win-win solutions. In zero-sum games this is not the case because the size of the "cake" is constant, so there are winners and losers. In both the general case and the zero-sum case, AI tools like the ones in this paper can help elevate less educated and less experienced players up to the same level as expert players, thereby making the distribution of value more fair.

A potential downside is that *if* the technology were only available to the privileged, that could increase unfairness.

## Acknowledgements

This material is based on work supported by the National Science Foundation under grants IIS-1718457, IIS-1617590, IIS-1901403, and CCF-1733556, and the ARO under awards W911NF1710082 and W911NF2010081.

## Footnotes

[1]Informally, the public game tree is the game tree visible to an observer with no knowledge of the players' private information.

[2]In the unbounded payoff case, the task is hopeless, since it is always possible for there to be a branch of infinite expectation that is reached so rarely that it has never been sampled.

[3]In the general-sum setting, finding an approximate Nash equilibrium is PPAD-complete, even for two players [30], so we do not hope to devise certificate-finding algorithms for that case.

[4]The pessimistic equilibrium is an equilibrium of the pseudogame. The optimistic profile may not be, hence the difference in naming.

[5]using either an exact method such as simplex, or an interior-point method such as barrier with crossover

[6]When the optimistic profile is not unique, the algorithm may waste work: for example, there may be one equilibrium which has support over pseudoterminal nodes and one which does not, the algorithm may pick the former and continue expanding nodes, making an unnecessarily big (but still correct) certificate.

[7]This can be easily formalized by perturbing by $\varepsilon$, then taking $\varepsilon$ sufficiently small. The strategy need not actually ever be constructed, so there is no need to formally discuss how small $\varepsilon$ needs to be; if coding this algorithm, we can simply store the indices of infinitesimal entries.

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
