[Supplementary Material]

# A Proofs

## A.1 Proposition 3.5

$$u_i^*(\sigma_{-i}) - u_i(\sigma) \le \beta_i^*(\sigma_{-i}) - \alpha_i(\sigma) \le \varepsilon. \qquad \square$$

## A.2 Proposition 4.1

By definition, it is impossible to reach any pseudoterminal node of $\tilde{G}$ by changing only a single player's strategy. Thus, for any player $i$, we have $\beta_i^*(\sigma_{-i}) - \alpha(\sigma) \le u_i^*(\sigma_{-i}) - u(\sigma) \le \varepsilon$. (the first inequality may not be an equality, because the best response $\beta_i^*(\sigma_{-i})$ is taken in the pseudogame, and $u_i^*$ is taken in the full game, where there is more flexibility. $\qquad \square$

## A.3 Theorem 4.5

**Lemma A.1.** *In every $\varepsilon$-NE of $G$, the entropy of P1's strategy is at least $T(1 - 2\varepsilon)$ bits.*

*Proof.* Let $\sigma_1$ be any P1 strategy in $\varepsilon$-equilibrium, and let $H_T$ be the entropy over terminal nodes when P1 plays $\sigma_1$ and P2 plays uniformly at random. Let $U_T$ be the number of rounds that P2 loses if she best responds to P1. Since $\sigma_1$ is an $\varepsilon$-NE strategy, we have $U_T \ge T(1/2 - \varepsilon)$. We will show that $H_T \ge 2U_T + T$, which will complete the proof.

Proceed by induction on $T$. For $T = 1$, the claim follows from the inequality $h(p) \ge 2\min(p, 1-p)$, which is true for all $p \in [0, 1]$, where $h$ is the binary entropy function.

In the inductive case, suppose that, at the top information set, P1 plays strategy $x = [p, q]$ (i.e. heads with probability $p$, and tails with probability $q$). Let $H' \in \mathbb{R}^{2\times2}$ be the matrix whose $ij$-entry is the conditional entropy over terminal nodes after P1 plays $i$ and P2 plays $j$ in the root information set. Similarly, let $U'$ be the matrix of conditional remaining expected number of rounds lost, not including this round, for player 2. Note that the utility matrix of the overall game, assuming that P2 plays correctly in later rounds, is $A := U' + I$. By IH, $H' \ge 2U' + T - 1$ element-wise. Further, P2's move in this information set does not affect the future of the game, since P1 does not learn P2's move, and P2's move does not otherwise affect her future optimal decisions. That is, $U'y$ is the same for all (normalized) $y$. Let $y$ be the uniform random strategy for player 1, and $y^*$ be a best response for player 1. Then we have:

$$\begin{aligned}
H &= 1 + h(p) + x^T H' y \\
&\ge T + h(p) + 2x^T U' y \\
&= T + h(p) + 2x^T U' y^* \\
&= T + h(p) + 2x^T A y^* - 2x^T y^* \\
&= T + h(p) + 2x^T A y^* - 2\min(p, 1-p)
\end{aligned}$$

and we are once again done by the inequality $h(p) \ge 2\min(p, 1-p)$. $\qquad \square$

The restriction on P2's strategy is necessary: indeed, since P1 has only $2^T$ pure strategies, there are sparse $\varepsilon$-NE strategies for P2 supported on only $O(T/\varepsilon^2)$ pure strategies.

Somewhat surprisingly, this proposition becomes false if P1 learns what P2 played in each round. Indeed, the P1 strategy "play heads if your number of losses minus number of wins is $\varepsilon T$, and uniformly at random otherwise" is (for large $T$) an $\varepsilon$-equilibrium with basically $T$ bits of entropy, since if P2 plays uniformly at random, with very good probability their score delta will never exceed $\varepsilon T$. However, despite having low entropy, this strategy has a very large support over terminal nodes.

**Corollary A.2.** *In every $\varepsilon$-NE of this game, for every $t \ge T/2$, the first $t$ rounds of P1's strategy have at least $t(1 - 4\varepsilon)$ bits of entropy.*

**Corollary A.3.** *Let $\varepsilon \le 1/16$. In every $\varepsilon$-NE of this game, for every $t \ge T/2$, P1's strategy assigns probability at least $2^{-t}$ to at least half of her pure strategies at round $t$.*

*Proof.* Let $Z$ be a random variable for P1's selected strategy, and $E$ be the event that $Z$ is among the half least likely pure strategies to be picked.

$$H(Z) = H(Z, E) = \Pr[E]H(Z|E) + \Pr[\neg E]H(Z|\neg E) + H(E) \leq \frac{2^t p}{2}\frac{t}{2} + \frac{t}{2}$$

where $H$ is the entropy. We know from above that $H(Z) \geq t(1 - 4\varepsilon)$, so the claim follows by solving for $p$. $\qquad\square$

We now prove Theorem 4.5. The proof acts like a partial converse to Proposition 4.1 for this game. Let $((\tilde{G}, \alpha, \beta), \sigma)$ be an $\varepsilon$-certificate, and let $Z'$ be the set of terminal nodes in $\tilde{G}$. Let $u$ be the assignment of utilities induced by P2 playing uniform random at every decision point outside $\tilde{G}$ (it does not matter at this point how P1 plays). Let $\sigma'_i$ be the uniform random strategy for player $i$. Then:

$$\beta_2(\sigma_1, \sigma'_2) \leq \beta^*_2(\sigma_1) \leq u_2(\sigma) + \varepsilon \leq u_2(\sigma'_1, \sigma_2) + 2\varepsilon = u_2(\sigma_1, \sigma'_2) + 2\varepsilon. \qquad\text{(A.4)}$$

For simplicity of notation, for any terminal node $z$ of $\tilde{G}$, let $r(z)$ be the number of rounds remaining in the game. Then note that $\beta(z) - u(z) = r(z)/2T$ for every $z$. Now suppose for contradiction that $\tilde{G}$ has fewer than $n := 2^{2T(1-16\varepsilon)-2}$ terminal nodes. Consider the level of the game tree after both players have made $t := (1 - 16\varepsilon)T$ moves; in other words, the level at which $r(z) = 16\varepsilon T$. This level has $4n$ nodes, so certainly $\tilde{G}$ must contain at most $1/4$ of the nodes at this level. Let $S$ be a set of half of the nodes of $G$ at level $t$ to which P1 assigns probability at least $2^{-t}$. Then $\tilde{G}$ contains at most half the nodes in $S$. Now observe that

$$
\begin{aligned}
\beta_2(\sigma_1, \sigma'_2) - u_2(\sigma_1, \sigma^*_2) &= \frac{1}{2T}\mathbb{E}_z\, r(z) \\
&\geq \frac{1}{2T}\sum_{z \in S\setminus\tilde{G}} \sigma_1(z)\sigma^*_2(z)r(z) \\
&\geq \frac{1}{2T}\frac{1}{2}2^{2t}2^{-t}2^{-t}r(z) = 4\varepsilon
\end{aligned}
$$

which contradicts (A.4). $\qquad\square$

## A.4 Theorem 4.2

We first introduce some terminology that will be useful in this section. The *realization plan* corresponding to a strategy $\sigma_i$ is the vector of reach probabilities $\sigma_i(s)$ for each *sequence* $s$ for player $i$. The constraints on valid realization plans are linear, and the payoff of a two-player zero-sum game can be expressed as a bilinear form $x^T A y$, where $x$ and $y$ are the realization plan vectors for the two players, and $A$ is a payoff matrix depending only on the terminal node values [20]. This bilinear program is known as the *sequence form* of a game.

**Lemma A.5.** *Let $x$ be any P1 strategy. Let $\hat{x}$ be a strategy profile defined by mixing uniformly at random over a multiset of $k$ independent sampled pure strategies from $x$, where*

$$k \geq \frac{D^2}{2\varepsilon^2}\log\frac{2N}{\delta}.$$

*and $D$ is the maximum support size over terminal sequences of any P2 pure strategy. Then with probability $1 - \delta$, for any strategy profile $y$, we have $|u_2(\hat{x}, y) - u_2(x, y)| \leq \varepsilon$.*

*Proof.* We follow basically the same idea as the proof in [26]. Let $A$ be the P2 sequence-form payoff matrix, restricted to those rows and columns corresponding to terminal sequences. By Hoeffding, we have

$$\Pr\Big[|(A\hat{x})_i - (Ax)_i| \geq \frac{\varepsilon}{D}\Big] \leq 2e^{-2k\varepsilon^2/D^2} \leq \frac{\delta}{N}$$

by picking $k$ as above. Taking a union bound over the at most $N$ sequences for P2, we have $\|A\hat{x} - Ax\|_\infty \leq \varepsilon/D$ with probability $1 - \delta$. Now select an $x'$ for which this is true. Then by Hölder's inequality, for any pure realization plan $y$, we have

$$\left|y^T A\hat{x} - y^T Ax\right| \leq \|y\|_1\|A\hat{x} - Ax\|_\infty \leq \varepsilon.$$

where the last inequality follows because $\|y\|_1 \leq D$. Now since $\left|y^T A\hat{x} - y^T Ax\right|$ is convex in $y$, and the pure realization plans are the vertices of the polytope of all realization plans, we are done. $\qquad\square$

Theorem 4.2 now follows by applying the lemma to an equilibrium strategy $x$ with any $\delta < 1$. ☐

## A.5 Theorem 5.1

Sampling this number of samples at each nature node $h$ is at least as good as sampling $(D^2/2\varepsilon^2)\log(2N/\delta)$ pure nature strategies. The proposition now follows by applying Lemma A.5 to the game in which the game tree is the same as $G$, P1 is nature, P2 controls every actual player in $G$ (and thus has perfect information), and the P2 utility function is $u$. ☐

## A.6 Corollary 5.2

By a union bound over the $|\mathcal{P}|$ players and the two utility functions $\alpha_i$ and $\beta_i$ for each player, we have that with probability at least $1 - 2\delta|P|$, for every $i$ and every deviation $\sigma_i'$, $|\hat{\alpha}_i(\sigma_i', \sigma_{-i}) - \alpha_i(\sigma_i', \sigma_{-i})| \leq \varepsilon$ and $\left|\hat{\beta}_i(\sigma_i', \sigma_{-i}) - \beta_i(\sigma_i', \sigma_{-i})\right| \leq \varepsilon$.

Let $\hat{\alpha}_i(\sigma)$ and $\hat{\beta}_i(\sigma)$ for a given strategy $\sigma$ be the utilities of $\sigma$ under the approximated version of $\tilde{G}$. Let $\hat{\sigma}_i^*$ be a best response for player $i$ in the approximated version of $\tilde{G}$, and let $\sigma_i^*$ be a best response in $\tilde{G}$ itself. Then we have:

$$\beta_i^*(\sigma_{-i}) \leq \hat{\beta}_i(\sigma_i^*, \sigma_{-i}) + \varepsilon \leq \hat{\beta}_i^*(\sigma_{-i}) + \varepsilon \leq \hat{\alpha}(\sigma) + \varepsilon + \varepsilon' \leq \alpha(\sigma) + 2\varepsilon + \varepsilon'$$

for every player $i$. ☐

## A.7 Proposition 6.3

Let $(x, y)$ be an $\varepsilon$-NE in the sense of Definition 6.2. Then

$$\beta^*(y) - \alpha(x, y) \leq \beta^*(y) - \alpha^*(x) \leq \varepsilon \quad \text{and} \quad \beta(x, y) - \alpha^*(x) \leq \beta^*(y) - \alpha^*(x) \leq \varepsilon. \qquad ☐$$

## A.8 Proposition 6.4

Let $(x, y)$ be an $\varepsilon$-NE in the sense of Definition 3.2. Then

$$\beta^*(y) - \alpha^*(x) \leq \beta^*(y) - \alpha(x, y) + \beta(x, y) - \alpha^*(x) \leq 2\varepsilon. \qquad ☐$$

## A.9 Theorem 6.5

We reduce from the SET-COVER problem, which is known to be NP-hard to better than a $\Theta(\log n)$ factor [29]. In SET-COVER, we are given a universe $U = \{1, \ldots, n\}$ and a collection of $m$ sets $\mathcal{S} = \{S_1, \ldots, S_m\}$ whose union is $U$, and our task is to find the smallest subset of $\mathcal{S}$ whose union is still $U$.

Consider the following game: P2 starts by choosing to either *play* or *leave*. If P2 leaves, then the game immediately terminates, and P1 gets value $1/2m$. If P2 chooses to play, then P1 chooses an index $i = 1, \ldots, m$. Then, P1 is given $m$ consecutive opportunities to leave the game (and immediately lose), should they choose. (The sole purpose of this is to inflate the size of the certificate.) After this, P2, without knowing the $i$, chooses an element $u \in U$. P1 gets value 1 if $u \in S_i$, and 0 otherwise.

This game has $\text{poly}(m, n)$ nodes, and its value (for P1) is exactly $1/2m$, since P1 can force P2 to leave by playing uniformly at random (and not choosing to lose). We now claim that, for $\varepsilon < 1/2m$, finding an $\varepsilon$-certificate of size $\Theta((m + n)k)$ is equivalent to finding a set cover of size $k$, which completes the proof.

If $\mathcal{R} \subseteq \mathcal{S}$ is a set cover of size $k$, then consider the trunk created by expanding exactly those P2 decision nodes where P1 has played some set $S_i \in \mathcal{R}$. This creates a trunk of size $\Theta((m + n)k)$. Even pessimistically, P1 can gain value $1/k \geq 1/m$ by randomizing uniformly over $\mathcal{R}$ in this trunk; thus, P2 is forced to leave, and this is a 0-certificate.

Conversely, suppose we had an $\varepsilon$-certificate, for $\varepsilon < 1/2m$, constructed from some tree $\tilde{G}$. Let $\mathcal{R}$ be the collection of sets $S_i \in \mathcal{S}$ for which P2's decision node after P1 plays $S_i$ has been expanded, and let $k = |\mathcal{R}|$. Then the trunk has size at least $\Omega((m + n)k)$. If $\mathcal{R}$ is not a set cover, then there is some $u \in U$ outside the union of sets in $\mathcal{R}$. If P1 plays $u$, then she gains optimistic value 0. Thus, since $\varepsilon < 1/2m$, $\mathcal{R}$ must be a set cover. ☐

## A.10 Theorem 6.6

Consider the family of two-player games in which there is a target string $x \in \{0,1\}^n$, and play proceeds as follows: Player 1 chooses, bit-by-bit, a string $y \in \{0,1\}^n$. If $x = y$, then Player 1 wins; otherwise, Player 2 chooses whether to win or lose. The smallest certificate in this game has size $\Theta(n)$, and consists of the path of play to $y$. However, there is no algorithm, randomized or deterministic, that will find the correct node $y$ without first expanding $\Omega(2^n)$ other nodes. □

## A.11 Theorem 6.8

($\Leftarrow$) Suppose $\tilde{G}$ has no 0-certificate. Let $(x^*, y_*)$ be an optimistic profile. Then

$$\alpha(x^*, y_*) \leq \alpha^*(y_*) < \beta^*(x^*) \leq \beta(x^*, y_*).$$

where the middle inequality is strict since $\tilde{G}$ has no 0-certificate, But then $\alpha(x^*, y_*) \neq \beta(x^*, y_*)$; i.e., there is some uncertainty as to the value of the strategy profile $(x^*, y_*)$; i.e., there is a nonzero probability that a pseudoterminal node is reached.

($\Rightarrow$) Now suppose $\tilde{G}$ has a 0-certificate, and call it $(x_*, y^*)$. Clearly, $(x_*, y^*)$ cannot contain in its support any pseudoterminal node. We claim that $(x_*, y^*)$ is also an optimistic profile of $\tilde{G}$, which completes the proof. Indeed, we have

$$\alpha^*(x_*) \leq \beta^*(x_*) \leq \beta^*(y^*) \quad \text{and} \quad \alpha^*(x_*) \leq \alpha^*(x^*) \leq \beta^*(y^*)$$

But all of these must actually be equalities, since $\alpha^*(x_*) = \beta^*(y^*)$ for a 0-certificate. Thus, $x_*$ is a Nash equilibrium strategy in $(\tilde{G}, \beta)$, and $y^*$ is a Nash equilibrium strategy in $(\tilde{G}, \alpha)$, which is what we needed to show. □

## A.12 Theorem 6.10

($\Leftarrow$) The correction algorithm adds infinitesimal amounts to sequences such that P2 is then forced to never play to any bad sequence that could be used to achieve value better than $V(I)$. Thus, corrected equilibrium is actually an $\varepsilon$-equilibrium for infinitesimal $\varepsilon$, and the proof of Appendix A.11 applies verbatim.

($\Rightarrow$) A pessimistic strategy will never be corrected, since a pessimistic player never has a terminal node of utility $+\infty$. Thus, again, the proof of Appendix A.11 applies verbatim. □