[Reviews · NeurIPS 2020]

Review 1

Summary and Contributions: This paper introduces the idea of a pseudogame with lower and upper bounds on leaves, based on the bounds on values of descedendant histories in a real game. It uses an equilibrium of these pseudogames as a certificate of equilibrium within a game. The authors show some problems that necessarily have a small certificate, an example showing not all games have a small certificate, and hardness results for finding a small certificate in a two player zero-sum game. They give an algorithm for finding a certificate in a two-player zero-sum game, (noting it might not terminate in reasonable time, given the hardness results) and run some experiments on a set of small games.

Strengths: The certificate idea is interesting, and an interesting extension of alpha-beta search bounds to more general class of extensive form games. The paper gives a fairly complete treatment, between the theoretical results and the experiments.

Weaknesses: The hardness results might limit the relevance to a narrower community: negative results are important, but a new idea that is immediately accompanied by an argument that it is in general hard to find will always be of interest to fewer people.

Correctness: Skimming the supplementary material, claims all appear to be correct. The experimental setup was reasonable.

Clarity: The paper was clear, with a reasonable layout.

Relation to Prior Work: The paper did a reasonable job of placing itself in the games domain. I am not aware of any other related work.

Reproducibility: Yes

Additional Feedback: Proposition 6.4 needs to flip Definition 6.2 and Definition 3.2. In the paragraph the section "Black-box setting", would it make more sense to use h rather than z for a node which might be non-terminal? The name optimistic equilibrium feels misleading by name, given it's not an equilibrium. Maybe optimistic profile? In the proof of theoremm 6.8, should the \alpha^*(y^*) be \alpha^*(y_*)? \alpha(x^*,y_*) <= \alpha^*(y_*) is still clearly true, and now \alpha^*(y_*)=\alpha^* which must be strictly less than \beta^* = \beta^*(x^*) -- comments after feedback The review comment about column generation being similar is good: it might be worth including in some way as related work.


Review 2

Summary and Contributions: This paper studies abstraction and the corresponding equilibrium certificate of extensive form games. The abstraction/pseudogame idea is intuitive, which is formed by merging some nodes in the extensive form tree into one meta-node with upper and lower bound representing the optimistic and pessimistic payoffs of all the possible outcomes. The overall goal of this paper is to find small certificates/epsilon-certificate of a huge extensive form game. The first part of the paper studies the properties of certificates. The authors connect the definition of epsilon-certificate of an extensive form game to the definition of epsilon-Nash equilibrium. The second part of the paper focuses on the cardinality of the small certificates. Due to the connection between epsilon-certificate and epsilon-Nash equilibrium, small certificates can be constructed by finding sparse epsilon-Nash equilibrium. According to the result from alpha-beta search in perfect-information zero-sum games with certain assumptions and in normal-form games, one can find the optimal solution through searching \sqrt{N} of the entire tree (with N nodes). This yields an efficient way to construct the psuedogame and the corresponding certificate (epsilon=0). However, in the general cases, the size of the certificates in extensive-form games can be quite large. The authors link the dependency of the cardinality of the certificates and the size of the information. They also show that in the worst case, the cardinality of the smallest certificate can be almost of the same order as the size of the nodes in the game tree. The third part of the paper assumes having only black-box access to the extensive form games. The black-box access is common but can jeopardize the randomness in the extensive form games. The authors provide analysis on sampling complexity in order to achieve equilibria with certain approximation error. Lastly, the authors focus on the two-player zero-sum extensive form games. They first provide a hardness result of finding the smallest certificate, which is NP-hard to compute any log N approximation in its size. However, they propose an iterative algorithm to find an small certificate. The idea is to compute the equilibrium in the current pseudogame, then expand the pseudo terminals to get a larger pseudogame and repeat. The algorithm terminates when there is no pseudo terminal that can be expanded to change the solution (i.e., the equilibrium is fixed). Therefore, the termination guarantees the optimality of the equilibrium. This idea is similar to standard column generation and branch-and-bound in operational research, and best-response algorithms in equilibrium finding problem. The authors also handle the cases when there are unbounded payoffs. They propose to refine the solution to avoid touching any bad sequences. It is achieved by slightly altering the computed solution with an infinitesimal amount to incrementally cover the support set. The authors also provide interesting theoretical analysis on three two-payer zero-sum extensive form games. The proposed algorithm can find small certificate compared to the original size of the entire games and approximated to the size of the minimal certificate.

Strengths: The paper studies extensive aspects of finding certificate in extensive form games. The proofs are sound, and the paper covers many hardness results of this problem. Along with the theoretical analysis, three experimental results are provided. The results are also convincing and consistent across all the experiments.

Weaknesses: One concern is the lack of theoretical novelty in this paper. Many of the techniques are similar to the existing concepts in game theory, e.g., alpha-beta search, column generation, and branch-and-bound algorithm. Hardness results in the paper are impressive, but others are relatively less surprising to me. Secondly, the proposed algorithm guarantees to find a small certificate, but there is no guarantee on the running time. The authors proposed some heuristics but didn’t provide an analysis on how much improvement can be achieved. My next concern is the application of small certificates. It sounds to me finding small certificates is significantly harder than solving extensive form games. But can small certificates be beneficial to solving extensive form games or be beneficial to any learning aspect in extensive form games? It would be great to have some empirical evidences showing that small certificates are actually helpful in other applied aspects. Similar to the last concern, the idea of exploitability was mentioned in the abstract and the introduction but never mentioned in the later part of the paper. If this is the main motivation of the paper, how do you define the exploitability and why does small certificate imply exploitability guarantee? How is this exploitability guarantee justified in your experiments? =========================================================== Thank you for your response. That indeed resolves my biggest concern about the small certificates. I have updated the score. I think the paper is a good paper despite most of the theoretical results are negative. It is still valuable to know how hard those problems are. One minor suggestion is to include a formal definition or at least a more detailed explanation of exploitability and small certificates. I was quite confused by the purpose of small certificates. It would help the readers to follow the purpose of the idea more.

Correctness: Yes, the claims are convincing, and the experiment methodology used in the paper is sound.

Clarity: Yes, overall it is well written. One minor point is that the structure of the paper can be improved to help the readers better understand the rich results of this topic.

Relation to Prior Work: Yes

Reproducibility: Yes

Additional Feedback: Typo in Proposition 6.4: Definition 6.2 and Definition 3.2 are misplaced.


Review 3

Summary and Contributions: The paper introduces a notion of small certificates for Nash equilibrium in extensive form games. The correctness of such certificate then can be verified in the time linear in the size of the certificate (rather than the game size). This is important as some games can be very large (even infinite) compared to the size of some certificate. Experimental section includes experiments on some medium-sized imperfect information games.

Strengths: I really enjoyed reading this paper. It reads well and has a nice logical flow. The core of the idea can be summarized as “generalizing alpha-beta idea to imperfect information settings / Nash equilibria”. Formalizing this via the idea of small certificates is a neat idea, and is a good contribution on its own. The paper goes much further than that, as it also includes non-trivial complexity results related to the algorithmic complexity. It is interesting that these results are rather negative. The paper also includes a simple iterative algorithm that produces certificates of (hopefully) small size. The experimental section is a nice bonus, as the paper would probably be also strong with just the theoretical results. It is also very cool that the results show a relatively small certificate for an infinitely sized limit Leduc. AFAIK the presented results are novel, significant and interesting to the community.

Weaknesses: Proposition 6.3: Any ε-NE in the sense of Definition 6.2 is an ε-NE in the sense of Definition 3.2. and Proposition 6.4. Any ε-NE in the sense of Definition 6.2 is a 2ε-NE in the sense of Definition 3.2. Is that just some issue of text merging? Table 1 should probably use “limit Leduc” rather than “Leduc” to make it more clear. It is not clear to me what results of section 6 are also relevant for perfect information. For example, Theorem 6.6: briefly looking at the proof, seems to hold perfect information games, while the proof of Theorem 6.5 seems to rely on imperfect information?

Correctness: All seem correct.

Clarity: The paper is well written and reads well.

Relation to Prior Work: AFAIK, the prior work is well cited. This is mostly due to the fact that there is very little previous work looking into small certificates in the settings of imperfect information games.

Reproducibility: Yes

Additional Feedback: Section 4.3 Extensive-form games with low information have small certificates, namely Theorem 4.2 deals with existence of equilibrium with small support sizes - relating that to the “how much information do the agents learn”. This section seems to be very closely related to the [1] “Schmid, Martin, Matej Moravcik, and Milan Hladik. Bounding the support size in extensive form games with imperfect information. Twenty-Eighth AAAI Conference on Artificial Intelligence. 2014.“. That work also relates “uncertainty” and support size in a similar way. Theorem 4.2 also only works for epsilon > 0, while you might be able to use [1] for results with exact Nash, as it presents bounds on small support sizes of exact Nash. ***************************************************************************************************************************************************************** Post Rebuttal: No change to my score - I thought this is a good paper and still think so. I appreciate the author's response to my questions.


Review 4

Summary and Contributions: This paper proposed a concept named \epsilon-certificate to guarantee the players' utilities at an \epsilon-Nash equilibrium of a game. In some cases finding the \epsilon-certificate only needs to explores a small portion of the game tree. These positive results utilize the analysis of the alpha-beta search and the (well-known) existence of \epsilon-Nash strategies with support logarithmic in the number of pure strategies. In addition to the positive results, a negative result shows that a small certificate does not always exist in extensive-form games. The paper then looks at black-box settings, where the game tree is not known in advance and can only be explored through some oracle. Some sample complexity results are presented. Then, two negative results follow: 1. no poly-time algorithm to approximate the smallest certificate within a logarithmic factor (by a reduction to the Set Cover problem). 2. no algorithm to solve a black-box zero-sum game in time subexponential w.r.t. the size of the smallest certificate. Finally, an LP based algorithm to find a certificate is presented. Some empirical results are provided to support the effectiveness of the algorithm.

Strengths: + An interesting concept, i.e., \epsilon-certificate, was proposed in the paper. + The motivation is convincing and the idea has practical significance. + The discussion of both positive and negative results seem sound. + A practical algorithm 6.7 is proposed to find a certificate in two-player zero-sum games in black-box settings. + Some empirical results are presented to support the theoretical predictions.

Weaknesses: 3.1 It seems that the positive result presented in section 4.2 assumes that a normal-form game can be converted to a perfect-information extensive-form game. This seems likely a misunderstanding, since obviously it isn't always true. What I am missing? **Note**: the rebuttal has clarified this point, which is no longer a concern. 3.2 The paper discusses "small certificate" in many places. It would be better to have a precise definition of "small" (e.g., sublinear) before the discussion. I was not sure what a small certificate is until question 4.4. 3.3 The sample complexity results in section 5 are interesting in order to deal with uncertain environment. However, they are not quite relevant to the remainder of the paper. 3.4 It would be better to have some insights for practical application from the hardness results in Theorem 6.5&6.6. Where does the hardness come from? Given a game, is it possible to control some parameter of the game such that the found certificate is provably small (e.g., size dependent on the parameter)? 3.5 An oracle is assumed to exist in black-box settings (line 169-173). Is this kind of oracle common in computer games or poker AI? What is the motivation for such an oracle? It would be more intuitive to have a concrete example for such an oracle. Post rebuttal: I have to admit that I am a little annoyed that the authors simply dismissed my main criticism of the paper (that it has what is ultimately redundant content). I do understand where they are coming from, and since I view this as a strong contribution nevertheless, it won't affect my recommendation. However, I suggest that the authors give more thought to the point: there is a way to narrate this without apparent redundancy. For example, they can provide a simple illustration to gain intuition (which they already do in essence with their first result), and then supply the general result. The paper would be even stronger if it had an experimental section in place of the redundant theory. Post rebuttal: I thank the authors for their effort in putting together a rebuttal. My view of the paper remains positive.

Correctness: The results seem sound, although 3.1 above needs some clarification.

Clarity: The paper is not difficult to follow. Some minor things can be improved (see 3.2 and 3.3).

Relation to Prior Work: The paper provides a sound connection to prior work.

Reproducibility: Yes

Additional Feedback:

[Author Response · NeurIPS 2020]

**Reviewer 1:** Thank you for the review! We agree with all your suggestions. We'll incorporate them. Thanks: thorough!

**Reviewer 2:** Thank you for the review!

*Novelty*: To our knowledge, this is the first approach that gives exploitability guarantees while only sampling a small portion of an extensive-form imperfect-information game. This is a highly novel and very important direction (we discuss importance further in the application paragraph below). These kinds of approaches have not been applied to extensive-form imperfect-information games before nor tested experimentally. Also, this approach is very different than the game abstraction literature (discussed in the introduction of the paper).

*Runtime guarantee*: Obviously, since the algorithm expands at least one node per iteration, the total number of iterations it takes is at most the total number of nodes in the full (underlying) game. You are correct, though, in saying that there is no general runtime guarantee for infinite games, nor one that is a (polynomial) function of the size of the smallest certificate. As we prove in the paper, these are impossible to achieve in the general case. However, we provide experimental evidence that our algorithm finds small certificates in a variety of games. We also want to emphasize that our run time is a function of the size of the certificate, not the size of the entire tree. This is key, and is in sharp contrast to all prior approaches, including CFR, MCCFR, EGT, etc.

*Application of small certificates*: This we justify in the paper. Despite general hardness, our algorithm allows us to find small certificates in practice of size much smaller than the size of the full game. The natural application (which we discuss in the paper) is one where the agent only has blackbox access to a large game. With the techniques in this paper, strategies with exploitability guarantees can be computed for such games. This is not possible with prior techniques in very large or even infinite games, as we explain in the paper. We demonstrate that even some infinite games have small certificates. It would be impossible to run most known algorithms on such an infinite game, because most of them require either expanding the whole game tree beforehand, or at least (in the case of, *e.g.*, outcome-sampling MCCFR) being able to bound the size of the game tree and conducting more samples than the number of leaves in the tree.

*Exploitability*: It is well known that exploitability in zero-sum games is bounded by the Nash gap ($\varepsilon$ throughout the paper), which we analyze extensively. We will remind the reader about this in the final version.

*Typo*: Yep, thanks! We'll fix that in the final version.

**Reviewer 3:** Thank you for the review!

*"Text merging" in Prop 6.4*: Typo: Prop 6.4 should have the two definitions flipped. We'll correct this in the final version.

*Limit Leduc*: We'll make this clarification in the final version.

*Perfect info in Sec 6*: Correct. And the dependence on imperfect information in Thm 6.5 is unavoidable: in perfect-info games, the smallest certificate can be computed from a game tree in linear time via an alpha-beta-like algorithm.

*Schmid et al. paper*: Thank you for the suggestion. We will try to obtain this paper (it is not currently available via aaai.org or any other site that Google can find) and will read it and cite it in the final version.

**Reviewer 4:** Thank you for the review!

(3.1) Section 4.2 doesn't assume perfect information. A normal-form game can be converted to a small *imperfect-information* game, as is described in that section. That is all that matters.

(3.2) Thank you for the suggestion. We will incorporate this into the final version.

(3.3) We include this section because it is sometimes unreasonable to assume direct access to the nature action distribution: in many black-box settings, we can only sample the nature distribution.

(3.4) As discussed in the response to Reviewer 3, the hardness in Thm 6.5 comes from the imperfect information, and is somewhat mitigated by the observation that we usually don't care about finding the *smallest* certificate, as long as we can efficiently find one of reasonable size. The hardness in Thm 6.6 is more fundamental: it comes from the fact that we're not assuming access to any reasonable heuristic of where to explore; thus, we may explore the optimal path of play last in the worst case, resulting in a large certificate. We'll include these summary sentences in the final version.

(3.5) Yes. 1) Upper and lower bounds on the value of a node: It is natural to assume trivial bounds (e.g., $[0, 1]$ or $(-\infty, \infty)$) on the utility of all nodes if we don't know any better. Often, we can do better. In our experiments, this was clear in goofspiel. More generally, often rewards are "incremental" (e.g., in a war game if one loses an asset, the value of the asset can be subtracted from the maximum payoff right then); in these cases, the bounds on deeper nodes are often much tighter than the trivial bound. 2) Player action list at player nodes: This seems reasonable. It is impossible to learn to play a game if we don't even know what we're allowed to do. 3) Nature samples at nature nodes: This seems reasonable. This is the minimum amount of access at nature nodes (those that are expanded) necessary for solving.

[Meta-Review · NeurIPS 2020]

The paper proposes novel concepts, epsilon-certificates and pseudo-games, for finding bounds on utilities of (approximate) Nash equilibria in game. Overall, the reviewers agree that this is a interesting novel contribution. The paper shows positive and negative results, but more negative than positive. This was a major point of discussion, but in the end, all the reviewers agreed that knowing these negative results is of important value to the community. Another question that arose was on the value of small certificates. This was largely cleared up by the author response and the discussion that followed. However, I encourage the authors to revise the paper based on these initial reactions, i.e. possibly move some arguments from the author response to the paper itself to clarify these points. There remain several issues pointed out by the reviewers that authors should address in their final version. Consider adding a citation to existing literature on bounding support sizes as pointed out by Reviewer #3. Also, the point of redundant theory made by Reviewer #4 is important and could affect the long-term impact.